# The Impact of Residual Dispersant on the Flocculation and Sedimentation of Synthetic Tailings in Seawater

**DOI:** 10.3390/polym14102085

**Published:** 2022-05-20

**Authors:** Rodrigo Yepsen, Leopoldo Gutiérrez, Pedro G. Toledo

**Affiliations:** 1Department of Metallurgical Engineering, Universidad de Concepción, Concepción 4030000, Chile; royepsen@gmail.com; 2Department of Chemical Engineering and Laboratory of Surface Analysis (ASIF), Universidad de Concepción, Concepción 4030000, Chile

**Keywords:** dispersants, HSMP, flocculation, sedimentation, chalcopyrite, muscovite, quartz, water quality, adsorption, salt bridge, polymer bridge, water cycle

## Abstract

Dispersants under certain conditions favor the flotation of molybdenite in seawater; however, it is not clear if the entrainment of residues to the thickening stage can compromise the quality of the clarified water. In this work, the impact of small concentrations of sodium hexametaphosphate (SHMP) on the flocculation and sedimentation of synthetic tailings containing kaolinite, muscovite, and quartz in seawater is evaluated. The flocculant polymer is a high-molecular-weight polyacrylamide, and the pH is alkaline. The results are auspicious for mineral processing. On the one hand, the impact of SHMP is not entirely negative and can be lessened by limiting entrainment, which is good for copper and molybdenum ore processing. On the other hand, if the small increase in turbidity generated by the SHMP is tolerated, it is possible to expect improved settling speeds. Without SHMP, large but light agglomerates are formed. With SHMP, smaller but denser aggregates are formed, settling faster, and minute aggregates increase turbidity. The underlying mechanism derives from the competition between SHMP and polymer chains for the cations in solution; the result is a greater repulsion between the chains, which leads to greater repulsion and thus dispersion of smaller flocculant coils. The study shows that SHMP in concentrations of 1 to 3 kg/t is perfectly acceptable. The results represent an advance in the understanding of SHMP interactions with polymers and minerals in water clarification, which should be of interest to the industry whose sustainability in some regions depends on closing the water cycle.

## 1. Introduction

Global water resources are becoming increasingly scarce due to climate change, making freshwater protection an urgent challenge for developing countries with arid climates. In Chile, for instance, the growth of the mining industry and the diversification of copper minerals require more water; therefore, proper water management, not only in the mining sector, is essential for the country’s sustainable economic development. The recovery and recirculation of good quality water from thickeners is key and is today mandatory for the sustainability of the industry [1,2]. The challenge is to recirculate as much water as possible, today between 70 and 80 percent, sufficiently clear and through a quick process [3,4]. Several conditions must be controlled, and the chemistry must be correctly chosen. However, even in minute amounts from upstream, unexpected factors such as chemical carryover can make the task extremely difficult. Seawater is a great alternative to continental water; in mineral processing, it is used raw, desalinated, or partially desalinated. Although seawater represents a natural solution to freshwater scarcity, its high ionic strength also brings adverse effects. High concentrations of different ions, as in seawater, induce important changes in the surface properties of copper molybdenum sulfides, strongly affecting flotation efficiency [5,6,7,8], and intervene in the interactions between copper sulfides and clay minerals. The depressant effect of clay minerals on chalcopyrite and molybdenite flotation is well known. The effect becomes stronger in the presence of most cations in seawater [9,10]. These cations induce degrees of hydrophilicity in these minerals, leading to low recoveries at pH > 9.5 even in the absence of clay minerals due to oxy-hydroxy complexes of Ca and Mg, which precipitate as Ca(OH)_2_ and Mg(OH)_2_ on the mineral surfaces [10,11]. These hydrolyzed species favor heterocoagulation between metal sulfide and clay mineral particles, explaining the low flotation efficiency [9].

The exploitation of finely disseminated low-grade copper sulfide ores is associated with several gangue mineralogical species, among which phyllosilicates are of particular interest because they critically interfere in flotation and thickening stages. Phyllosilicates include serpentites, pyrophyllites, clays, micas, and chlorites. In particular, micas and clays, have three major negative effects: inhibit collector adsorption and the formation of particle-bubble electrostatic bridges [12,13,14,15], increase the viscosity of pulps [16], and hinder the tailings thickening process [17]. Of interest in this study are the clay kaolinite and the mica muscovite, which are the most common gangue minerals and, together with quartz, make up the most recurrent tailings composition. The electrical charge and spatial distribution that these minerals develop on their surfaces in water as a function of pH determine their adsorption preferences. Kaolinite is a clay mineral with the formula Al_2_Si_2_O_5_(OH)_4_. It is a layered silicate mineral with one tetrahedral sheet of silica linked through oxygen atoms to one octahedral sheet of alumina [18,19,20,21]. The basal siloxane plane is the least reactive surface of kaolinite; it is hydrophobic and thus repels water molecules. One possibility of having a charge on this plane is by substituting Al^3+^ and/or Fe^3+^ for Si^4+^ in the tetrahedral layer, which produces a small and permanent negative charge in this basal plane; this charge is structural because it does not depend on pH [22]. On the basal gibbsite plane, each hydroxyl group coordinates with two aluminum atoms. The dependence on pH is weak, but with a remarkable feature, it would have a pH of zero charge, under which this basal surface is cationic and above which is anionic (around 7 according to Gupta et al. [22]). The edges of kaolinite carry a charge that depends on the solution pH [22,23,24,25,26,27]. According to the literature, alumina surfaces are generally positively charged at pH < 8.7, and silica surfaces are positively charged only at ultra-acidic pH, pH < 2–3; therefore, it can be concluded that the positive charge at the edges of kaolinite originates from gibbsite-like border sites. Gupta et al. [22] determined that the isoelectric point of the edges occurs at pH between 4.5 and 5. At alkaline pH, the basal siloxane plane remains uncharged; instead, the basal gibbsite plane and the edges acquire a negative charge. Through molecular simulation, Quezada et al. [28,29,30,31] showed that the edges of kaolinite at alkaline pH and above the pH of zero charge become negatively charged with surface groups, including two AlOH^−1/2^ and one SiOH per unit cell. The formula for muscovite is usually given as KAl_2_(AlSi_3_O_10_)(OH)_2_, although small amounts of other elements commonly substitute for the more typical constituents. Like all mica minerals, muscovite is a layered phyllosilicate with a structure consisting of layers bonded together by potassium cations. Layers are of two types (for instance, see Kurganskaya and Luttge [32]). A tetrahedral shell consists of silicon or aluminum cations, each surrounded by four oxygen anions forming a tetrahedron, with three of these oxygens shared with neighboring tetrahedra to form a hexagonal sheet. The fourth oxygen is apical. Additionally, an octahedral shell consists of aluminum cations, each surrounded by six oxygen or hydroxide anions, forming octahedrons. These octahedrons share anions to form a hexagonal sheet similar to sheets made up of tetrahedra. These layers are firmly bound through the apical oxygens of the tetrahedral sheets. Potassium atoms are located in the structure in large cavities between layers, providing a weak bonding, which in turn provides muscovite with a characteristic perfect basal cleavage that leads to two reactive planes—basal and edges. The edges expose silanol, aluminol, and acidic hydroxide groups, making them more reactive and dependent on pH than the basal surfaces, more than three hundred times according to [33,34]. The basal surface is much less reactive and only carries a structural charge due to the isomorphic substitution of lattice elements, which does not depend on pH. The zeta potential of muscovite, dominated by the edge charge, is positive at acidic pH ca. 2, is zero at pH around 3, defining the isoelectric point, and is permanently negative at pH > 3, reaching a maximum at pH between 8 and 10 [35]. Quartz, the second most abundant mineral on earth, is a silicate made from pure silica. The atoms are bonded in a continuous framework of silicon-oxygen tetrahedrons of SiO_4_. Each oxygen is shared between two tetrahedra, giving a general formula of SiO_2_. Quartz in water is capable of adsorbing and releasing protons, a behavior attributed to the amphoteric character of its surface silanol groups. Thus, these groups can exist in deprotonated (Si-O^−^), unprotonated (Si-OH), and protonated (Si-OH^2+^) states. Silanol groups are partially protonated below the pH corresponding to the point of zero charge and deprotonated above that point. As the pH increases, the number of deprotonated silanol groups and the negative charge on the surface increases. As the pH decreases, the number of protonated silanol groups and the positive charge on the surface increases. The point of zero charge for silica occurs at pH ~ 2 [36,37].

The thickening stage requires a fine choice of polymer flocculants, usually polyacrylamides, so that at the alkaline pH that prevails in mineral processing, the phyllosilicate and quartz particles form dense aggregates that settle quickly. Opposing mechanisms arise in seawater. The cations cover the mineral surfaces and facilitate the formation of salt bridges with the polymeric chains of the flocculant. However, the cations are also adsorbed on the anionic groups of the flocculant, which without repulsion between their groups, winds up forming small skeins of very short range, thus losing its main flocculant characteristic. Which mechanism dominates depends on the system considered and the prevailing conditions. Cationic flocculants effectively minimize turbidity, although the aggregates formed are dense and difficult to transport. Anionic flocculants are preferred.

Dispersants are widely used in mineral flotation to modify colloidal interactions and thus prevent particle aggregation [38]. Among the dispersants, sodium hexametaphosphate (SHMP), whose chemical formula is Na_6_[(PO₃)_6_], is the most widely used [6,7]. SHMP can adsorb on cationic sites at the edges of clay particles, allowing the surface charge of the clay to be controlled by their anionic sites. In saline solutions, another possibility is that the SHMP forms complexes with cations desorbed from the mineral surface, exposing its anionic charge. The result is always a negative surface charge on the particles and thus the electrophoretic mobility, which increases the repulsive forces between them [39]. A critical aspect related to the sustainability of the flotation and thickening processes is the impact of the SHMP fractions transported from the flotation stage to the water clarification stage. A recent study shows that the addition of SHMP under certain conditions favors the flotation of molybdenite in seawater [6]. An evaluation of the impact of the SHMP in the thickening stage requires estimating how it alters the mineral surfaces and the type and strength with which it interacts with the flocculant polymer chains in freshwater or seawater, as appropriate. These two elements define the quality of the water to be recirculated, and the type of aggregates that ultimately determines the amount of energy and water necessary for its transport to its final disposal.

This work evaluates the impact of small concentrations of SHMP entrained from upstream processes on the flocculation and sedimentation of synthetic tailings containing kaolinite, muscovite, and quartz in seawater. The flocculant polymer is a high-molecular-weight polyacrylamide, and the pH is alkaline. The results represent an advance in the understanding of SHMP interactions with polymers and minerals in water clarification in thickeners, which should be of interest to the industry whose sustainability in some regions depends on closing the water cycle.

## 2. Materials and Methods

### 2.1. Materials

Samples of muscovite and kaolinite used in this work were obtained from Ward’s Natural Science Establishment, which were manually dry ground to −2 mm and kept in sealed plastic bags. pH was adjusted using lime in all the tests. Sodium hexametaphosphate (SHMP) obtained from Sigma Aldrich (97% purity) was used as dispersant. Seawater from the coast of the BíoBío Region was used; the chemical composition varies slightly from place to place, hence the concentrations in Table 1 are assumed. The flocculant was CYFLOC A-150 HW, a polymer containing anionic polyacrylamide functional groups, which is referred to as LPAM.

### 2.2. Settling Rates and Turbidity

In these tests, the liquid medium is seawater. The influence of SHMP on the settling properties was assessed by settling tests in a 2-L glass cylinder using the LPAM flocculant in dosages of 10, 20, 30, 40, and 50 g/t. SHMP concentrations were between 0 and 5 kg/t. First, the tailings suspensions were homogenized for 15 s in the cylinder, and then the position of the interface between pulp and liquid as a function of time was registered for 30 min. Finally, the sedimentation rate was obtained from the slope of the initial linear section of the registered curve. Turbidity measurements of the supernatant samples taken after 6 min of settling were taken using a HACH 2100N turbidimeter. The artificial tail was made up of 15% phyllosilicate (muscovite or kaolinite) and 85% quartz, both by mass. The artificial tail was first crushed in a cone crusher to a particle size of −2 mm and then wet ground using carbon steel balls as grinding media to obtain a P80 of 150 μm in a laboratory ball mill at 67% solids content.

The solutions of flocculant were prepared daily following a procedure described elsewhere [41,42]. Accordingly, stock solutions at 0.35 g/L were prepared at 20 °C by mixing 175 mg of LPAM with 500 mL of a 0.01 M NaCl solution in a beaker under magnetic agitation for 5 h. The beaker was covered with a parafilm and with an opaque lead to prevent exposure to light and degradation. The stock solutions were kept for a maximum of 12 h, after which they were discarded.

### 2.3. Electrophoretic Mobility and Zeta Potential

To better understand the effect caused by SHMP on the electrophoretic mobility of kaolinite and muscovite particles, measurements were performed using a Zetacompact Z9000 from CAD Instruments. The suspensions were prepared by mixing 0.07 g of particles less than 20 µm in size in aqueous calcium and magnesium solutions. Then, the suspensions were conditioned for 3 min under magnetic stirring. The ions in the solutions strongly compress the electrical double layer and neutralize the charges on the surface of the particles; thus, in these tests, the ionic strength had to be kept to a minimum with calcium and magnesium concentrations as low as 0.05 M. In these tests, sodium hydroxide was used instead of lime as a pH modifier to avoid divalent calcium ions.

The zeta potential of quartz was measured in seawater with different concentrations of SHMP in a pH range between 2 and 11. The tests were carried out in the Zetacompact Z9000. Again, low concentrations of calcium and magnesium had to be used.

### 2.4. Flocculant Adsorption

The flocculant adsorption on muscovite, kaolinite, and quartz was evaluated separately by measuring the Total Organic Carbon (TOC) in their respective suspensions. A Shimadzu TOC-L was used, which works by oxidizing the flocculant chains to carbon dioxide, the value of which was converted into TOC measurements.

In these experiments, 20 g of each mineral sample were mixed and dispersed for 15 min in 150 mL of flocculant solutions at different concentrations (10, 20, 30, 40, and 50 g/t) in seawater at pH 9 and different doses of SHMP (0, 1, 2, 3, 4, 5 kg/t). The suspensions were then centrifuged, and the clarified water samples were analyzed for TOC values. The specific adsorption was calculated in mg of flocculant adsorbed per g of mineral (mg/g).

### 2.5. Flocculant Viscosity

The intrinsic viscosities of flocculant solutions in seawater were measured at SHMP concentrations between 0 and 5 kg/t. The flocculant polymer doses were between 0.001 and 0.006 g/DL. Always, solution pH was 9. Viscosity measurements were made with the viscometer Cannon-Fenske 200 capillary, cleaned before each test with distilled water and acetone to remove contaminants. In these experiments, LPAM solutions previously conditioned to the required pH, concentration, and mechanical degradation were transferred to the capillary viscometer reservoir enclosed in a water bath to maintain the temperature at 25 ± 0.10 °C.

## 3. Results

### 3.1. Settling Rate

Figure 1 shows the effect of the SHMP dispersant dose on the sedimentation rate of phyllosilicate tailings made up of muscovite or kaolinite at 15% (*w*/*w*) and quartz at 85% (*w*/*w*) in seawater. The tests with muscovite and quartz show that the addition of SHMP between 1 and 3 kg/t slightly increases the sedimentation rate at all doses of flocculant tested between 10 and 50 g/t; however, the highest values are obtained at the highest dosage of flocculant. Between 3 and 5 kg/t of SHMP decreases the settling rate for any dose of LPAM. In short, 1 kg/t SHMP and 50 g/t LPAM lead to the highest settling rate. On the other hand, the tests carried out with kaolinite and quartz reveal that the settling rate increases with SHMP until it reaches a maximum value, and then decreases to a stationary value. However, unlike the behavior of the muscovite-quartz system, the SHMP dose effect is not monotonic with the LPAM dose. At any dose of SHMP, the highest value of settling rate is obtained at 20 g/t of LPAM. The highest settling rate value is obtained with SHMP at 2 kg/t and LPAM at 20 g/t. The results indicate that the sedimentation rate depends on the relationship between the SHMP and LPAM doses. Judging by these results, SHMP could have a positive effect on the sedimentation stage of tailings containing muscovite and kaolinite.

### 3.2. Turbidity

The turbidity of muscovite-quartz suspensions (15/85% *w*/*w*) in seawater at pH 9 changes non-monotonically with SHMP concentration for all LPAM doses tested. It also changes non-monotonically with LPAM dose at a fixed SHMP concentration (Figure 2a). The cloudiest supernatant is obtained at 3 kg/t SHMP and 30 g/t LPAM and the clearest at 1 kg/t SHMP and 50 g/t LPAM. The turbidity of kaolinite-quartz suspensions (15/85% *w*/*w*) in seawater at pH 9 increases monotonically with an increase in SHMP concentration for any LPAM dose (Figure 2b). It also changes monotonically with the dose of LPAM for a fixed concentration of SHMP. The cloudiest supernatant is obtained at 5 kg/t SHMP and 10 g/t LPAM and the clearest without SHMP and 50 g/t LPAM. The presence of SHMP in the pulps increases the turbidity as expected, even at the lowest concentrations. However, it is also clearly observed that an increase in the flocculant dose causes a significant decrease in turbidity, in agreement with previous investigations [42].

The turbidity of each mineral in aqueous suspension is of interest to determine the most affected by the dispersant, thought initially to disperse clays. Figure 3 shows turbidity results for suspensions of kaolinite, muscovite, and quartz at different doses of SHMP and flocculant at pH 9 in seawater. Quartz is the most affected, even at the lowest concentrations of SHMP. At intermediate doses of SHMP (1 to 3 kg/t), the turbidity decreases with the dose of LPAM, as expected. Muscovite turbidity is also affected by SHMP concentration, but only at high concentrations; at 5 kg/t SHMP the turbidity is almost as high as in quartz suspensions at the same SHMP concentration. At low SHMP (between 1 and 3 kg/t), the turbidity of muscovite practically does not change. Remarkably, the concentration of LPAM has a little differentiating effect on this clay. The turbidity of kaolinite suspensions is notably lower compared to muscovite and quartz suspensions. The turbidity increases monotonically with SHMP concentration for all LPAM doses tested and decreases monotonically with LPAM doses.

### 3.3. Flocculant Adsorption

The various minerals considered in this study show different sedimentation and turbidity values and trends in the presence of SHMP in the pulps. Therefore, it is also of interest to determine if the differences also exist in flocculant adsorption. For this, TOC tests were performed.

Figure 4 shows flocculant adsorption per mineral mass for pure muscovite, kaolinite, and quartz in seawater at pH 9. For all these minerals in suspension at a fixed dose of flocculant, the adsorption of the polymer, as the concentration of SHMP changes, is relatively constant, except for quartz at SHMP concentrations between 3 and 5 kg/t and for muscovite at the highest concentration of SHMP, where flocculant adsorption decreases.

The effect of SHMP in reducing flocculant adsorption follows the order quartz > muscovite > kaolinite. In other words, SHMP definitely interferes with the flocculant function in the quartz suspension. However, such interference in muscovite and kaolinite suspensions is not so evident. We are aware that the measurements of flocculant adsorption inferred from measurements of carbon in the supernatant of a suspension are affected by the lifetime of the mineral-polymer contact and that reaching equilibrium is difficult, so these results are valued more as a trend than as absolute values.

### 3.4. Electrophoretic Mobility and Zeta Potential

Figure 5 shows the electrophoretic mobility distributions of kaolinite and muscovite particles at pH 9 in aqueous solutions of Ca and Mg at concentrations as low as 0.05 M in the absence and presence of SHMP. The effect of SHMP on the electrophoretic rate is considerably more pronounced in the presence of Mg than Ca. In the presence of Mg, the effect of SHMP is to increase the negative value of the electrophoretic mobility of muscovite and kaolinite. This is because the SHMP entertains the Mg cations more than the Ca ones, forming some type of complexes and revealing the negative charge of both clays, which increases particle repulsion and thus increases dispersion, the ultimate purpose of SHMP. In these tests, no flocculant was used because the concentration of SHMP was too low to interact with the flocculant chains, mainly because the objective was to decipher the dispersive mechanism of SHMP.

It was also of interest to evaluate the dispersing effect of SHMP on quartz particles in an aqueous solution with 0.05 M Ca and Mg, separately, through the Zeta potential of the particles. Figure 6 shows the results with and without SHMP (1 g/t). In the entire pH range studied, the zeta potential of quartz is negative, showing that the low concentration of cations does not effectively neutralize the surface charge of quartz. When SHMP is added, both with Ca and Mg (0.05 M each), at the indicated low concentration, the Zeta potential becomes increasingly negative. In the same argument as above, SHMP forms complexes with Ca and Mg, decreasing the surface charge screen of quartz. Thus, the net effect of the SHMP is a greater dispersion of the particles.

### 3.5. Flocculant Reduced Viscosity

Reduced viscosities of CYFLOC A-150 HW flocculant at different doses of SHMP at pH 9 in seawater are shown in Figure 7. The reduced viscosity, the specific viscosity divided by the flocculant dose, decreases with flocculant dose for a fixed concentration of SHMP and decreases also with SHMP concentration for a fixed flocculant concentration. For a fixed concentration of SHMP, the reduced viscosity decreases as the flocculant dose increases. High concentrations of flocculant and SHMP lead to the lowest reduced viscosities.

## 4. Discussion

The adsorption of specific polymeric flocculants on mineral surfaces in aqueous solutions leads to the aggregation of solid particles and their separation from the liquid by sedimentation. The objectives are twofold and include clarified water to be returned to the process and solid aggregates with adequate size and density for cost-effective transportation to their final destination in tailings dams. The choice of process chemistry is further complicated if the water is of low metallurgical quality, such as seawater or partially desalinated water, and if chemicals other than flocculant are carried into the thickening units. Therefore, it is particularly interesting to evaluate the presence of SHMP, used in copper-molybdenum plants, in the flocculation and sedimentation of kaolinite clay, muscovite mica, and quartz, the main components of tailings. The chosen polymer flocculant is a polyacrylamide containing acrylamide monomers (>CHCONH_2_) intercalated with acrylate monomers (>CHCOO^−^). At the working pH (pH 9), all the mineral surfaces of interest are anionic, and the chosen flocculant is completely ionized; that is, all the –COOH groups are charged in the form of –COO^−^. The anionicity of the minerals and the flocculant chains is considerably reduced in seawater, the cations coat the mineral surfaces and form complexes with the flocculant; however, polymer adsorption does not occur exclusively through weak van der Waals forces. Recent molecular dynamics simulations show that at alkaline pH in seawater or water with high ionic strength, polyacrylamides are adsorbed on minerals, preferentially forming salt bridges [29,30,43], which are electrostatic interactions between the oxygens of the mineral surfaces and the (charged) oxygens of the acrylate groups of the polymer mediated by seawater cations, leading to mineral-cation-acrylate bridges. To a lesser degree, adsorption also occurs through hydrogen bonds between the oxygens on the mineral surfaces and the uncharged oxygens of the flocculant acrylamide groups, leading to mineral-cation-acrylamide bridges. The simulation results have shown that polyacrylamide adsorption on kaolinite, specifically on the more reactive edges, is much higher than on quartz. Using the statistical description of adsorbed polymers in terms of simple conformations such as loops, trains, and tails [44,45,46], Quezada et al. [29,30] have shown that adsorption on kaolinite occurs in loops and tails mainly, however, in quartz adsorption occurs almost exclusively as tails.

In the absence of SHMP, the sedimentation rates improve with the dose of flocculant; however, there is an optimal dose as shown in Figure 1. For the muscovite-quartz system, the optimal dose is between 40 and 50 g/t, while for the kaolinite-quartz system is between 20 and 40 g/t. An increase in sedimentation rates at higher doses is not necessarily the product of improved flocculation. Turbidity is not affected by the flocculant dosage for muscovite-quartz, as shown in Figure 2a. It is lowest for kaolinite-quartz at the optimal dosage of 40–50 g/t, shown in Figure 2b. As expected, flocculant adsorption on each mineral increases with dose, as shown by the experimental results in Figure 3. At a fixed dose of flocculant, the adsorption is somewhat higher on muscovite and kaolinite than on quartz, which agrees with the trend shown by the simulation results, although not as higher. These adsorption differences between individual minerals, especially clays versus quartz, impact differently when the minerals are mixed. Therefore, the optimal dose for sedimentation (Figure 1) is not the dose at which the individual minerals show the greatest adsorption of flocculant (Figure 3). Thus, the optimal dose of flocculant for the sedimentation of the muscovite-quartz system is the highest (50 g/t). However, the optimum for the kaolinite-quartz system is not the highest (20–40 g/t). Therefore, the highest dose of flocculant (50 g/t) in this last system leads to lower settling rates (Figure 1). This is the reference thickening and clarification behavior from which to evaluate the presence of variable amounts of SHPM that are entrained from upstream in mineral processing. The delicate mineral-polymer arrangement in a highly saline environment, such as seawater, can be easily broken by an exogenous component such as SHMP, even in minimal quantities.

In the presence of SHMP, the results change significantly. At the optimal flocculant dose for muscovite-quartz pulp (50 g/t), the sedimentation rate increases by 20%, from 2.5 cm/min to 3 cm/min, at SHMP concentrations between 1 and 3 kg/t. On the other hand, at the optimum flocculant dose for kaolinite-quartz pulp (20 g/t), the sedimentation rate increases 110%, from 1 cm/min to 2.1 cm/min, at SHMP concentrations between 1 and 3 kg/t. These results suggest that (anionic) SHMP in concentrations between 1 and 3 kg/t improves the sedimentation rate of clay-quartz pulps, especially when the flocculant is applied at an optimal dose. Flocculant doses of 50 g/t are unusual in plants; doses of 10 g/t and lower are common. The results in Figure 1 show that although settling rates are lowest at 10 g/t flocculant, the presence of SHMP at concentrations between 1 and 3 kg/t still improves the rate. Therefore, it can be concluded that the presence of SHMP between 1 and 3 kg/t improves the sedimentation rate of clay-quartz systems at any dose of flocculant, at least between 10 and 50 g/t. The question is whether the improvement in settling implies improved flocculation efficiency.

The turbidity of the supernatant in the two philosillicate-quartz systems could help to reveal the synergy between flocculant and dispersant. Considering that the clarified water from industrial thickeners has turbidity that varies between 50 and 100 NTU, it can be considered that the turbidity of the supernatant in the muscovite-quartz system is low. Even in the presence of SHMP, it is comparatively lower at the dose of flocculant, which maximizes the settling speed (50 g/t). At SHMP concentrations between 1 and 3 kg/t SHMP, the turbidity increases slightly and could be tolerated if the SHMP shows that it improves the performance of the flocculant. Turbidity of the supernatant in the kaolinite-quartz system is also low, although not as low as in muscovite-quartz. With SHMP, turbidity increases to industrial limits of 100 NTU. At the flocculant dose that maximizes the settling speed (20 g/t), the turbidity increases from 70 to 85 NTU when the SHMP concentration varies between 1 and 3 g/t. Without SHMP, the turbidity is only 30 NTU. Again, the increase in turbidity in the presence of SHMP could be tolerated if the SHMP shows that it does not threaten the flocculation-sedimentation process.

The turbidities of muscovite-quartz and kaolinite-quartz suspensions in SHMP are much lower than the turbidity of suspensions of the pure mineral components (Figure 3). However, the results in Figure 3 are still helpful for understanding the results in the mineral mixtures. In the absence of SHMP, the clarified water turbidity of clay-quartz pulps at any flocculant dose is comparable to the clarified water turbidity of the pure mineral pulps. However, if analyzed in more detail, the turbidity of kaolinite-quartz is slightly lower, and that of muscovite-quartz pulps is slightly higher. However, in the presence of SHMP, especially at concentrations greater than 3 kg/t, the turbidity of clay-quartz pulps is considerably lower than that of quartz alone, with a tendency to decrease with SHMP concentration (Figure 2). At first sight, the high turbidity of the supernatant in quartz pulps is striking, which increases by an average of 100 NTU, and in muscovite pulps, which increases by an average of 150 NTU when the concentration of SHMP increases from 3 to 5 kg/t (Figure 3). However, it should not be surprising because the anionic dispersant acts by removing seawater cations adsorbed on the mineral particles, preventing them from doing so and thereby increasing the repulsion between particles and decreasing the anchoring flocculant chains; this behavior is common to all doses of flocculant. The impact of the dispersant on kaolinite is considerably less; the turbidity increases on average by about 40 NTU when the concentration of SHMP increases from 3 to 5 kg/t (Figure 3).

The origin of the lower turbidity of the clay-quartz pulps compared to pure minerals in the presence of the dispersant has yet to be explained. An analysis of the flocculation behavior of the different minerals may be the key. Clays are very good adsorbents of almost everything, especially cations; they are defined as water structure makers [37,47,48]. At alkaline pH, they are strongly anionic—a condition shared with polyacrylamide-type flocculants. The flocculant chains are anchored to the clay surfaces, preferentially forming salt bridges, which, according to molecular simulation, may not be permanent [43]. On the other hand, quartz is not a great adsorbent of cations, and if it has any preference, it is not for highly charged cations; it is defined as a weak water structure maker or simply as a water structure breaker [37,47,48]. At alkaline pH, polyacrylamide chains do not have a strong preference for quartz, and without adsorbed cations, it is impossible to form salt bridges [30,43]. In the few anchor points on the quartz, the flocculant chains offer their tails to become entangled with others and thus form aggregates. This flocculation mechanism that is not mediated by cations is known as polymeric bridging. In this way, in the flocculation of clay-quartz pulps, the flocculant forms dense networks of salt bridges that trap quartz particles, which contribute to flocculation through a few polymeric bridges. The formed aggregates settle, leaving a supernatant with relatively low turbidity, which is lower than pure quartz pulps. When the pulp contains SHMP, even in low concentration, the strongly anionic phosphates at alkaline pH bind with the cations of the seawater, whether they are free in solution or desorbed from the minerals. Two effects are confronted. On the one hand, the anionic charge of the minerals is exposed, which increases the repulsion between the particles and thus the turbidity. On the other hand, the flocculant chains can unwind and act a little more freely. The increase in turbidity of the clay-quartz pulps when SHPM increases at all flocculant doses in Figure 3 suggests that the first effect dominates.

The flocculant adsorption results shown in Figure 4 for pure mineral pulps are not very different and do not vary significantly as a function of SHMP concentration. Adsorption increases with flocculant dosage on all minerals as expected. However, if analyzed more closely, the flocculant adsorbs somewhat less on the quartz than on the clays. As a trend, it agrees with the simulation results, but the adsorption on quartz is expected to be lower. The measurements in Figure 4 are based on monitoring the carbon footprint in the supernatant. As such, the amount of carbon varies strongly over time. What is recorded in Figure 4 is the amount of flocculant immediately after the experiment, which means that there is no time for sedimentation. However, if time is allowed to pass, the carbon content also increases by adsorption from atmospheric air. Figure 4 is the best that can be measured, and therefore it is valued that it shows that the flocculant is adsorbed somewhat less in quartz than in clays.

So far, it can be concluded from Figure 1, Figure 2 and Figure 3 that the presence of SHMP in concentrations of 1 to 3 kg/t is fully compatible with flocculants based on polyacrylamides at pH 9 or higher.

Seawater contains three times more magnesium than calcium (Table 1). However, both create greater problems in mineral processing than the predominant sodium, especially at the highly alkaline pH. They form hydrophilic hydroxides that coat the valuable minerals, impairing their flotation efficiency [9,11]. Adding to the latter is forming complexes with the flocculant chains that wind up and thus deactivate. The electrophoretic mobility and the related Zeta potential of colloidal particles are severely affected by ionic strength, which screens or weakens the net charge of the particles, reducing the extent of electrostatic forces within the colloidal sample. Therefore, it is not practical to measure these properties in seawater. However, it is still of interest to evaluate the effect of the addition of SHMP on the charge of mineral particles in the presence of key electrolytes of seawater such as Ca and Mg, which are known to be complex to handle in mineral processing. Figure 5 shows electrophoretic mobility data of kaolinite and muscovite particles, and Figure 6 shows Zeta potential data of quartz particles, all at pH 9 in aqueous solutions of Ca and Mg 0.05 M and traces of HSMP (ca. 1 ppm). Clays are strongly anisotropic with well-defined faces and edges and, therefore, the electrophoretic velocity cannot strictly be converted into Zeta potential as it can be done for quartz. The dispersant effect is more pronounced in the presence of Mg than in Ca. Both the electrophoretic mobility of muscovite and kaolinite are strongly shifted towards more negative values. In addition, the Zeta potential of quartz is markedly more negative. The dispersant prefers Mg more than Ca to form complexes because the former is smaller and more electronegative. As a consequence, the anionic charge on the mineral surfaces is more exposed, which possibly limits the anchoring of flocculant chains, which in turn would explain the increase in turbidity (Figure 3) and the restricted flocculant adsorption (Figure 4) even if the flocculant dose increases (Figure 4). The conditions of the suspensions in these tests are not those of process-relevant pulps but are those allowed in mobility measurements; however, the results of these limiting cases support the above assumptions.

The intrinsic viscosity in the presence of SHMP is of interest because it represents the contribution of the flocculant to the solution (solvent) viscosity, which is altered by the presence of SHMP. The intrinsic viscosity could be calculated by extrapolating reduced viscosity to zero flocculant concentration. We do not extrapolate here, but clearly, the trends in Figure 7 show that intrinsic viscosity decreases as SHMP concentration increases indicating that the polymer becomes smaller at higher SHMP. The intrinsic viscosity measures the interaction between the flocculant and the surrounding liquid; thus, the data in Figure 7 show that the SHMP effectively disfavors such interaction. As SHMP and flocculant chains compete for the cations in solution, the result is a greater repulsion between the chains, which leads to greater repulsion and thus dispersion of smaller flocculant coils. The photomicrographs in Figure 8 summarize the implications of the SHMP in water clarification. The SHPM disperses the flocculant chains, preventing their self-association.

A great concern is the accumulation of SHMP in the final link of the process, which is precisely the water clarification stage. For this reason, somewhat high SHMP concentrations were used, in the range of 1 to 5 kg/t, knowing that the optimum concentration in molybdenite flotation in seawater is only 0.15 kg/t [6]. This study does not address the environmental impact of the SHMP carried in the tailings, which is a great pending task.

The results of this study are auspicious for mineral processing in several ways. On the one hand, the impact of SHMP is not entirely negative and can be lessened by limiting entrainment to thickening units, which is good news for copper and molybdenum ore processing that benefits from dispersants. On the other hand, if the increase in turbidity generated by the SHMP is tolerated, it is possible to expect improved settling speeds; in any case, the increase in turbidity does not exceed the usual field NTU. Without SHMP, large but light agglomerates are formed. With SHMP, smaller but denser aggregates are formed, which settle faster and tiny aggregates that increase turbidity. The challenge is faster sedimentation with high turbidity (although with acceptable plant values) or slower sedimentation with minimal turbidity. The study shows that SHMP in concentrations of 1 to 3 kg/t is perfectly acceptable. Thus, the use of SHMP should not be discouraged. Another focus for improvement is the choice of flocculants, an inescapable issue because fines appear and affect more and more, and this will continue whether or not dispersant is used.

## 5. Conclusions

The impact of dispersants, in particular sodium hexametaphosphate (SHMP), on the flocculation and sedimentation of synthetic tailings containing kaolinite, muscovite, and quartz in seawater at alkaline pH is not entirely negative and can be lessened by limiting entrainment to the thickening units, which is good news for copper and molybdenum ore processing that benefits from dispersants. Without SHMP, large but light agglomerates are formed. With SHMP, smaller but denser aggregates are formed, which settle faster, as well as tiny aggregates that increase turbidity. The challenge is faster sedimentation with high turbidity (although with acceptable plant values) or slower sedimentation with minimal turbidity. The underlying mechanism stems from competition between SHMP and polymer chains for cations in seawater; the result is an increased repulsion between the polymer chains, which intensifies the repulsion and thus the dispersion of smaller flocculant coils. The study shows that SHMP in concentrations of 1 to 3 kg/t is perfectly acceptable. Settling rates and flocculant adsorption increase and viscosity decreases at the expense of a slight and tolerable increase in turbidity. Therefore, the use of SHMP should not be discouraged.

## Figures and Tables

**Figure 1 polymers-14-02085-f001:**
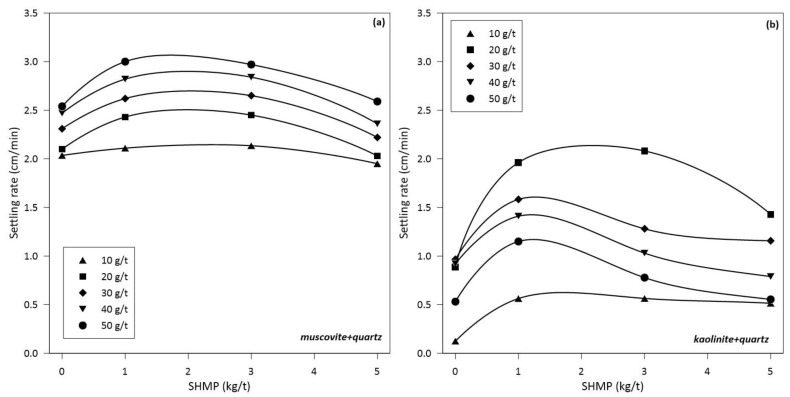
Sedimentation rate of phyllosilicate flotation tails (30% solids) as a function of SHMP dosage in seawater at pH 9 at different flocculant concentrations (in g/t). (**a**) 15% (*w*/*w*) muscovite and 85% (*w*/*w*) quartz, (**b**) 15% (*w*/*w*) kaolinite and 85% (*w*/*w*) quartz.

**Figure 2 polymers-14-02085-f002:**
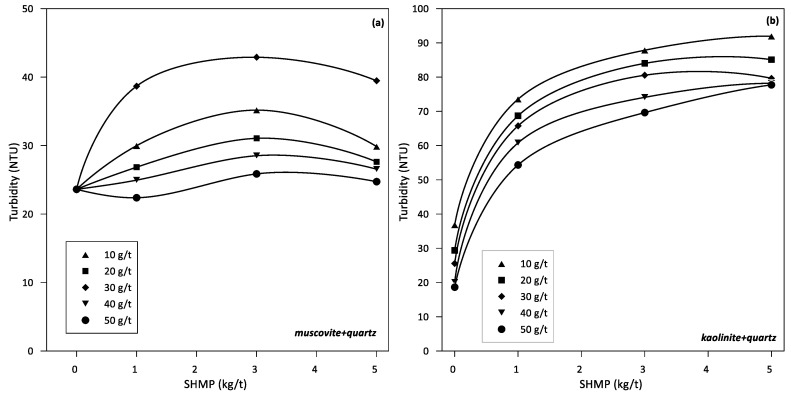
Supernatant turbidity of phyllosilicate flotation tails (30% solids) as function of SHMP concentration in seawater at pH 9 at different flocculant concentrations (in g/t). (**a**) 15% (*w*/*w*) muscovite and 85% (*w*/*w*) quartz, (**b**) 15% (*w*/*w*) kaolinite and 85% (*w*/*w*) quartz.

**Figure 3 polymers-14-02085-f003:**
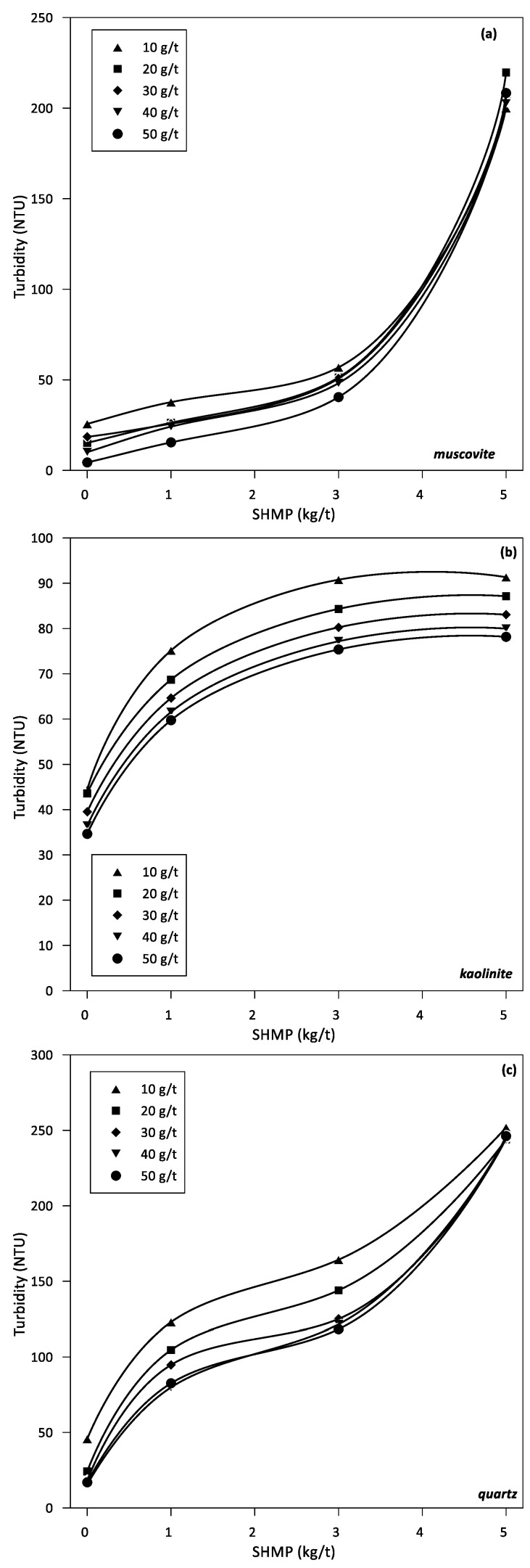
Supernatant turbidity of muscovite (**a**), kaolinite (**b**), and quartz (**c**) in seawater at pH 9 (30% solids) as function of SHMP concentration in seawater at pH 9 at different flocculant concentrations (in g/t).

**Figure 4 polymers-14-02085-f004:**
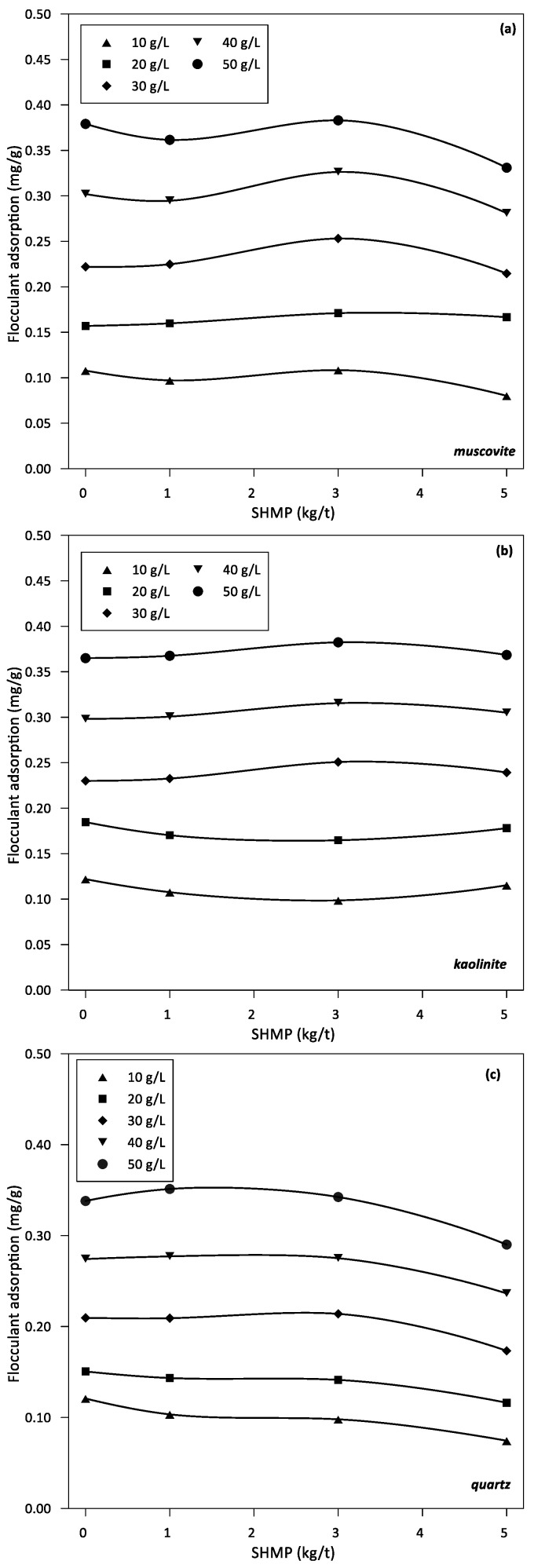
Flocculant adsorption versus SHMP concentration for muscovite (**a**), kaolinite (**b**), and quartz (**c**). pH 9 in seawater. Flocculant dose in g/L.

**Figure 5 polymers-14-02085-f005:**
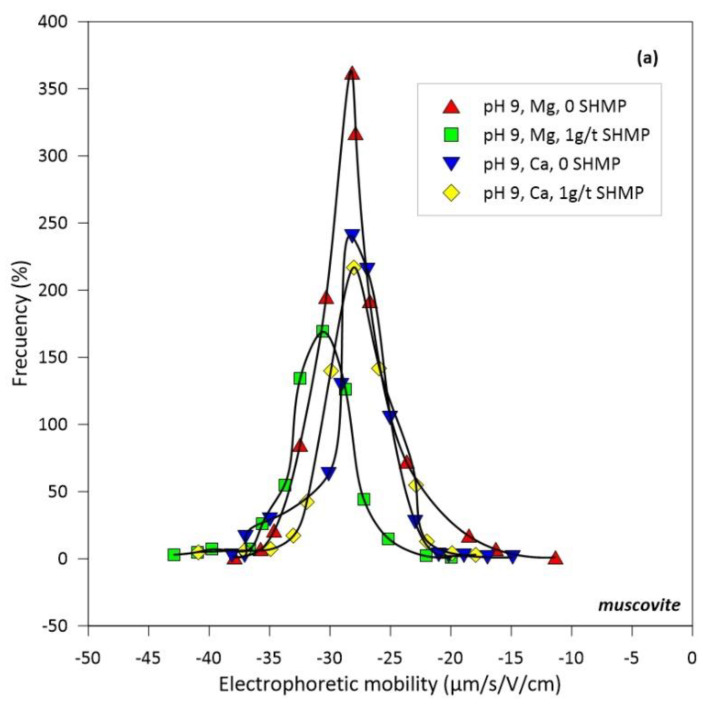
Electrophoretic mobility for muscovite (**a**) and kaolinite (**b**). pH 9, effect of Mg (0.05 M), Ca (0.05 M), and SHMP (1 g/t).

**Figure 6 polymers-14-02085-f006:**
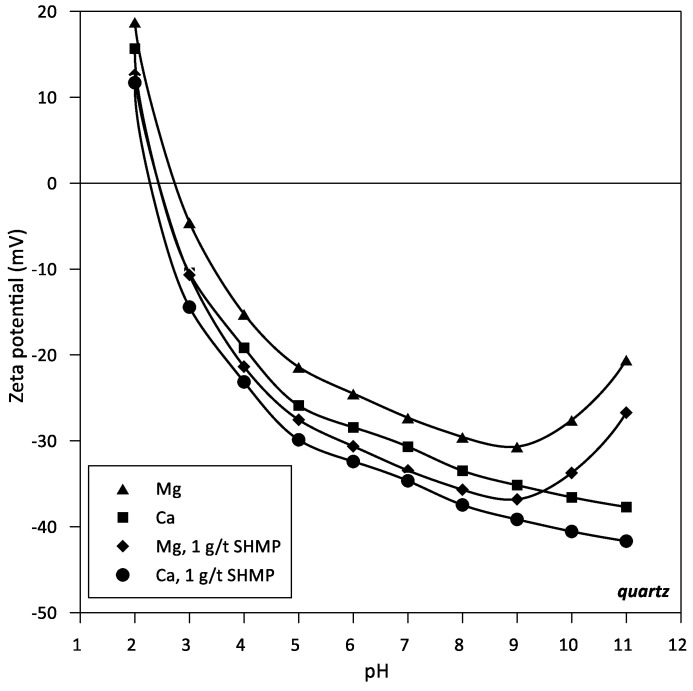
Zeta potential of quartz suspension in Ca and Mg solutions (0.05 M each). 1 g/t SHMP.

**Figure 7 polymers-14-02085-f007:**
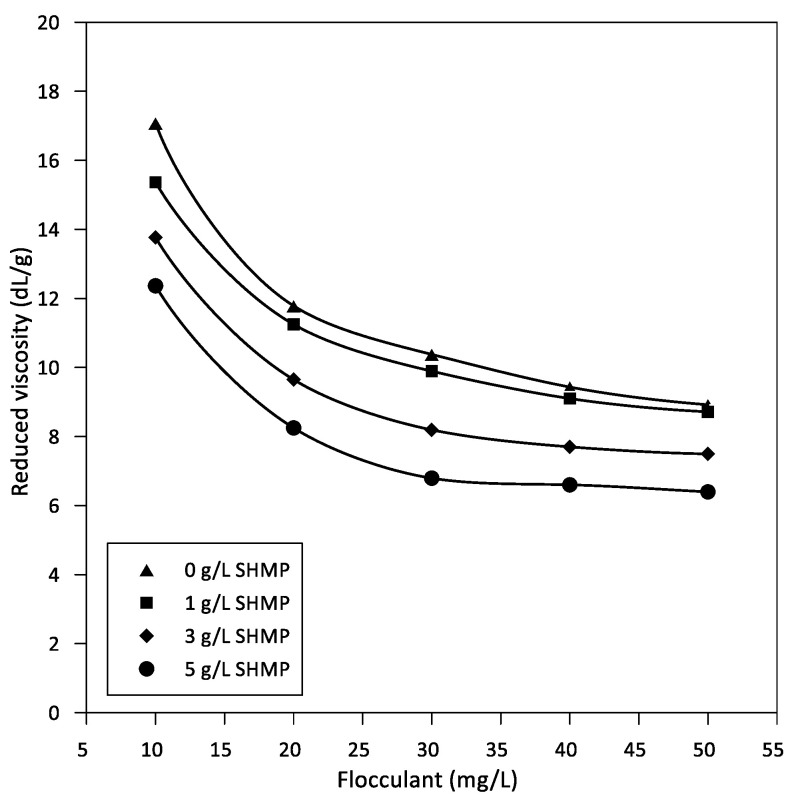
Reduced viscosity of CYFLOC A-150 HW flocculant as a function of flocculant dose at different SHMP concentration.

**Figure 8 polymers-14-02085-f008:**
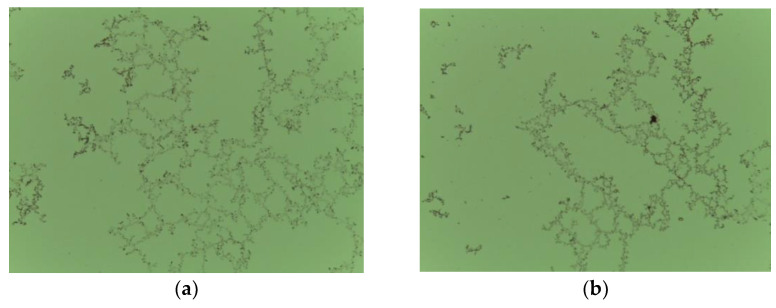
Photomicrographs of flocculant chains fully expanded without SHMP (**a**) and fragmented in small flocculant coils in the presence of SHMP (**b**), both at pH 9 in seawater.

**Table 1 polymers-14-02085-t001:** Ions and concentrations in seawater [40].

Ions	Concentration, mg/L
Cl^−^	19.345
Na^+^	10.752
SO_4_^2−^	2.701
Mg^2+^	1.295
Ca^2+^	0.416
K^+^	0.390
HCO_3_^−^	0.145
Br^−^	0.066
BO_3_^3−^	0.027
Sr^2+^	0.013
F^−^	0.001
Others	<0.001

## Data Availability

The data presented in this study are available on request from authors R. Yepsen and L. Gutiérrez.

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
