# Peer review of "The Impact of Residual Dispersant on the Flocculation and Sedimentation of Synthetic Tailings in Seawater"

_polymers, 2022, doi:10.3390/polym14102085_

Round 1

Reviewer 1 Report

The authors of this manuscript (MS) have examined the effects of SHMP residue on the flocculation and sedimentation of synthetic flotation tailings in thickeners and quantified the effectiveness of thickener operation using a series of measurements such as settling rate, supernatant turbidity, Toc tests for flocculant adsorption on mineral surfaces, Electrophoretic mobility rate, Zeta potential and viscosity. The authors have also found that the range of 1 to 3 kg/t of SHMP concentration in the tailings was optimal to achieve an acceptable turbidity of supernatant and faster settling rate of sediments. This reviewer believes that the investigation is thorough, and conclusion is convincing. More importantly, this MS prompts a novel thinking about flow sheet design of mineral processing because traditionally the tailings were given after an optimal flotation processing. In the future, perhaps, we should seriously take the efficiency of tailings processing into account when designing minerals flotation processing including the types and quantities of flotation reagents. Therefore, this paper is worth to be published in Polymers.

To further improve the clarification and presentation of this MS, this reviewer would like to make the following suggestions for the authors’ considerations.

  • There is a big block of texts in the “Discussion” section. It is difficult to read even though the paragraphs were logically written. It would be great if authors can insert a few schematics to help readers to quickly understand what the authors want to disseminate. For example, SHMP influences in supernatant (clarified water) and sediments (big agglomerates and small aggregates) and competition between SHMP and LPAM chains for cations in seawater.
  • The perfectly acceptable SHMP quantity in the tailings was 1 to 3 kg/t, or 1000 to 3000 ppm. Is it a reasonable concentration as a residue? In the reference 40, the use of dispersants in flotation of molybdenite in seawater was up to 150 ppm. Please explain the discrepancy and the reason of having such a high concentration range.
  • Figure 5 (a) and (b) are hard to read due to closeness of Mg and Ca curves in the individual tailing. The curves would be separate nicely if authors plot Mg in Muscovite and Kaolinite together as (a) and Ca as (b) because the EM values of ion Mg or Ca are quite different in Muscovite and Kaolinite tailings.
  • There were a few places that flocculant (noun) and flocculent (adjective) were misused. For instance, on line24, “… smaller flocculant coils” should be … smaller flocculent coils.”.

Author Response

Many thanks to the reviewer

Reviewer 2 Report

The introduction is too long, there is no need for that. Feel free to shorten it a bit.

Line 127: ..is the most widely used. (put some ref. where were used).

Which metode were used to calculate chenmical compositions? Explain

The discussion is extensive and follows the results. The conclusion is concise and corresponds to the results.

Add a couple of newer references  with the same / similar issues

Author Response

Many thanks to the reviewer
